# Association between attending exercise-based cardiac rehabilitation and cardiovascular risk factors at one-year post myocardial infarction

Ingela Sjölin[1,2], Maria Bäck[3,4], Lennart Nilsson[5], Alexandru Schiopu[1,2], Margret Leosdottir[1,2]*

1 Department of Clinical Sciences Malmö, Faculty of Medicine, Lund University, Malmö, Sweden,
2 Department of Cardiology, Skane University Hospital, Malmö, Sweden, 3 Division of Physiotherapy, Department of Medical and Health Sciences, Linköping University, Linköping, Sweden, 4 Department of Occupational Therapy and Physiotherapy, Sahlgrenska University Hospital, Gothenburg, Sweden, 5 Department of Medical and Health Sciences, Linköping University, Linköping, Sweden

* margret.leosdottir@med.lu.se

**Data Availability Statement:** The data used in this study is from the SWEDEHEART registry. Access to data from the registry needs to be applied for

## Abstract

### Background

Randomized trials confirm the benefits of exercise-based cardiac rehabilitation on cardiovascular risk factors. Whether exercise-based cardiac rehabilitation provides the same favourable effects in real-life cardiac rehabilitation settings, in the modern era of myocardial infarction treatment, is less well known. We examined the association between attending exercise-based cardiac rehabilitation and improvements in cardiovascular risk factors at one-year post myocardial infarction in patients included in the Swedish heart disease registry, SWEDEHEART.

### Methods

In this retrospective registry-based cohort study, we included 19 136 patients post myocardial infarction (75% men, 62.8±8.7 years) who were registered in SWEDEHEART between 2011 and 2013. The association between attending exercise-based cardiac rehabilitation (43% participation rate) and changes in cardiovascular risk profile between baseline and one-year follow-up was assessed using multivariable regression analysis adjusting for age, comorbidities and medication.

### Results

Attenders more often reported to have stopped smoking (men 64% vs 50%; women 64% vs 53%, p<0.001 for both, only smokers at baseline considered), be more physically active (men 3.9±2.5 vs 3.4±2.7 days/week; women 3.8±2.6 vs 3.0±2.8 days/week, p<0.001 for both) and achieved a slightly larger reduction in triglycerides (men -0.2±0.8 vs -0.1±0.9 mmol/L, p = 0.001; women -0.1±0.6 vs 0.0±0.8 mmol/L, p = 0.01) at one-year compared to non-attenders. Male attenders gained less weight (+0.0±5.7 vs +0.3±5.7 kg, p = 0.01) while

and third-party data usage is not allowed, irrespective of whether the data contain potentially identifying or sensitive data or not. Instead, given ethical study approval from the Swedish Ethical Review Authority, access to SWEDEHEART data supporting the present findings can be applied for from the Uppsala Clinical Research Center (UCR) in Sweden at datauttag@ucr.uu.se. Further information can be found on the UCR https://www.ucr.uu.se/en/ and Swedish Ethical Review Authority https://etikprovningsmyndigheten.se/ websites.

**Funding:** The authors received no specific funding for this work.

**Competing interests:** The authors have declared that no competing interests exist.

female attenders achieved better lipid control (total cholesterol -1.2±1.4 vs -0.9±1.4 mmol/L, p<0.001; low-density lipoprotein -1.2±1.2 vs -0.9 ±1.2 mmol/L, p<0.001) compared to non-attenders.

## Conclusions

In an unselected registry cohort of patients post myocardial infarction, compared to non-attenders those attending exercise-based cardiac rehabilitation achieved significantly larger improvements in cardiovascular risk factors at one-year after the acute event.

## Introduction

Comprehensive cardiac rehabilitation (CR) for patients with coronary artery disease (CAD) is provided through an interdisciplinary approach and includes specific core components such as risk factor management with cardio-protective medication and behavioural modification, patient education, psychosocial interventions, physical activity counselling and exercise training [1]. International guidelines consistently identify exercise training, generally referred to as exercise-based CR (exCR), as a cornerstone of comprehensive CR [2]. Randomized trials and observational studies including patients with myocardial infarction (MI) have confirmed the benefits of exCR in terms of reductions in cardiovascular mortality and re-hospitalization [3], cardiovascular risk factor management [4] and improved aerobic capacity [5]. Consequently, exCR has received the highest possible (IA) recommendation in the latest European Guidelines on Cardiovascular Disease Prevention [2].

However, many studies on exCR were performed before statins, angiotensin converting enzyme (ACE) inhibitors and percutaneous coronary interventions (PCI) became an integral part of MI treatment and often predominantly included male patients, limiting generalizability to both sexes [6, 7]. Whether the benefits of exCR on patient outcomes apply in the modern era of MI treatment and to men and women alike has in recent years been questioned [8]. Also, it is unclear whether attendance in exCR results in the same benefits in real-life settings as that observed in randomized trials, especially so in women, as several studies have indicated both uptake and outcomes in CR to be inferior among women [9–12].

The aim of this study was to examine whether attending exCR as a part of comprehensive CR is associated with beneficial changes in cardiovascular risk factor levels between admission and one-year follow-up in patients post MI from a real-life setting in the modern era of MI treatment. For this purpose we used data from the Swedish Web-system for Enhancement and Development of Evidence-based care in Heart Disease Evaluated According to Recommended Therapies (SWEDEHEART) registry.

## Materials and methods

### Patient population and settings

In this retrospective registry-based cohort study, we included all available patients registered in SWEDEHEART with an MI diagnosis (ICD code I21) during 2011, 2012 and 2013, and who thereafter attended a one-year registry follow-up visit within CR (n = 19 136) (Fig 1).

In Sweden, it is advised that all patients with suspected acute MI should be included in the SWEDEHEART registry during hospitalization. In 2011–2013 95% of all Swedish hospitals attending to patients with acute MI reported to SWEDEHEART, and compared to the Swedish

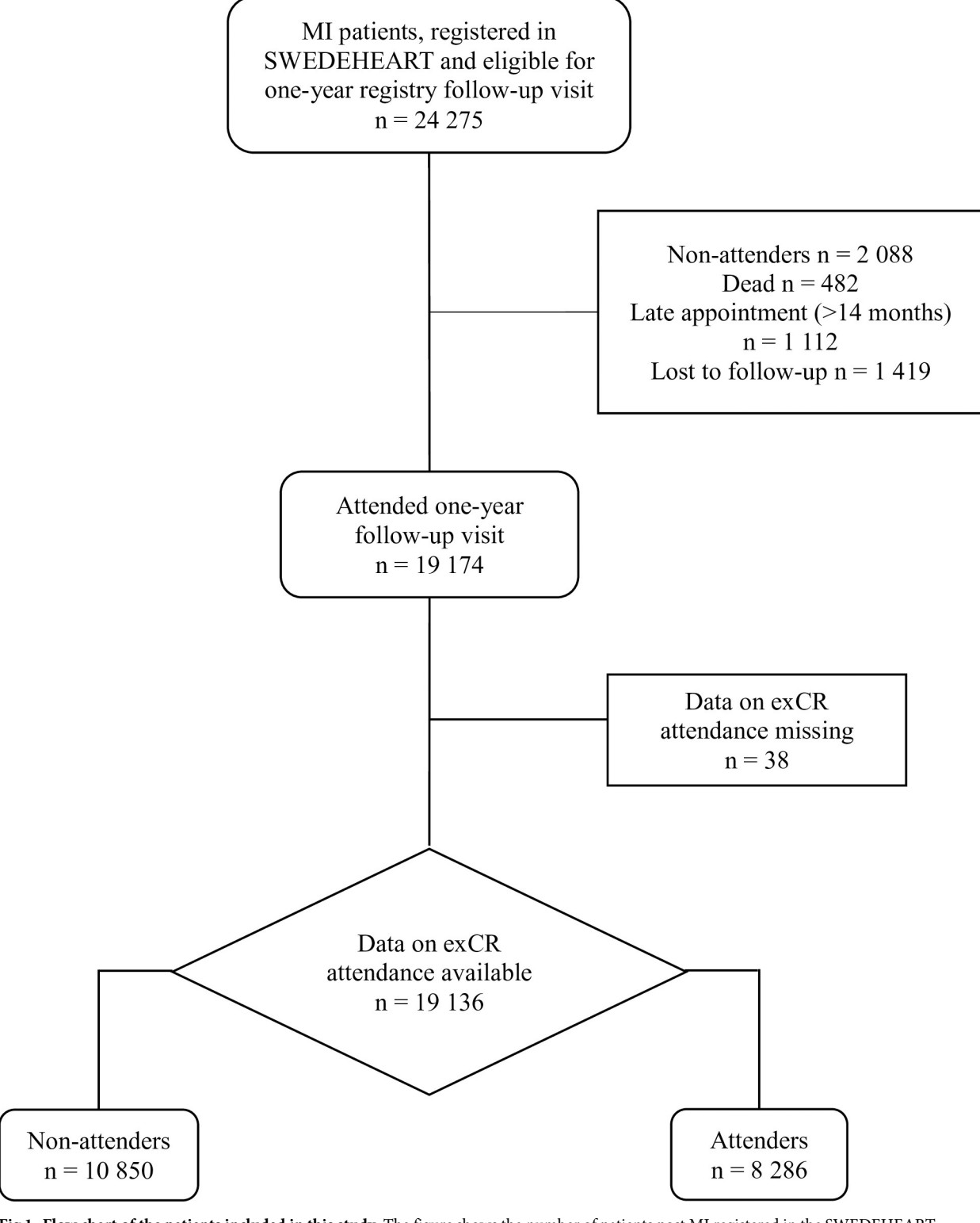

**Fig 1. Flow chart of the patients included in this study.** The figure shows the number of patients post MI registered in the SWEDEHEART registry in 2011–2013 who attended one-year follow-up thereafter and where data on exCR attendance was available.

National Diagnosis Registry the SWEDEHEART registry's median coverage for patients under the age of 80 years who had been hospitalized for acute MI was 91% in 2011 and 92% 2013[13, 14]. The registry includes more than 100 variables describing the acute care of MI, as well as approximately 80 variables describing performances in CR [15]. Baseline characteristics and details of the acute MI treatment are collected by healthcare personnel during hospitalization. Follow-up data describing treatment provided during CR, recurrent events and patient out-comes are collected at two routine follow-up visits at the CR centres at two-months (time frame 6–10 weeks) and one-year (time frame 12–14 months) after the index event. In 2011–2013 it was obligatory to register patients <75 years of age, while registration of those 75 years or older was optional. Approximately 90% of Swedish CR centres were represented in SWE-DEHEART in 2011–2013 and one-year follow-up data covered 75–80% of all patients <75 years of age who were alive one year after the MI [13, 14].

## Study variables

The exposure variable in this study was attendance (yes/no) in a structured exCR programme during the first year post MI (self-reported by the patient or documented by a physiotherapist in the patient´s medical records), as reported at the one-year follow-up visit. The definition of an exCR programme according to SWEDEHEART is a supervised centre-based programme, led by a physiotherapist, containing both aerobic exercise and resistance exercises with a total session length of 60 minutes. Each exercise session should include at least 20 minutes of aero-bic exercise (intensity 13–17 on Borg´s rating of perceived exertion (RPE) scale) [16] and resis-tance exercise performed in 1–3 sets of 8–10 upper and lower limb exercises in 10–15 repetitions. The programme should include three sessions per week whereof one session can be conducted at a healthcare facility or at home. While it is recommended that the duration of the exercise programme should be at least three months, the number of completed exCR ses-sions was not documented in SWEDEHEART at the time of the study. Patients are recom-mended to start exCR as soon as possible after discharge, however, due to limited physiotherapy resources at many CR centres, the most common starting time at exCR was approximately four weeks post-MI.

Other registry variables used in this study are summarized in Table 1.

**Disease history.**   Previous history of CAD (MI, percutaneous coronary intervention (PCI) or coronary artery by-pass grafting (CABG)), heart failure, diabetes, hypertension or stroke is registered at hospital admission (self-reported).

**Employment status.**   At hospital admission and at both follow-ups visits, the patient is asked about whether he/she is employed, unemployed, on sick leave, retired or studying/other.

**Medication.**   Current medication is registered at admission and at discharge from the hos-pital, as well as at both follow-up visits.

**Smoking status.**   The patient is asked if he/she is a current smoker, previous smoker (stopped smoking >1 month) or never smoked, at hospital admission and at both follow-up visits.

**Physical activity.**   The patient reports at both follow-up visits how many days during the last week he/she has been physically active for a minimum of 30 minutes (at least 10 minutes at a time) becoming short of breath and having a slightly increased pulse, corresponding to a brisk walk.

**Height and weight.**   During hospitalization the patient´s height (cm) is registered (mea-sured or self-reported). The weight (kg) is measured if possible, otherwise it is self-reported. Weight is measured at both follow-up visits. Body mass index (BMI) was consequently calcu-lated (kg/m$^2$).

**Table 1. Variables from SWEDEHEART used in this study.**

| Variable name | Definition/units | Hospitalization | One-year follow-up |
|---|---|---|---|
| Disease history | Prior history of diabetes, CAD, heart failure or hypertension | x | |
| Employment status | Employed, unemployed, on sick-leave, retired or studying | x | |
| Medication | Medication for diabetes, CAD, heart failure or hypertension | x | x |
| Smoking status | Never smoked, previous smoker or current smoker | x | x |
| Physical activity | Days of at least 30 minutes of physical activity during the last week (0–7) | | x |
| Height | Measured in cm or self-reported | x | |
| Weight | Measured in kg or self-reported | x | x |
| Body Mass Index | Calculated (kg/m$^2$) | x | x |
| Blood pressure | mmHg | x | x |
| Fasting plasma glucose | mmol/L | x | x |
| Total cholesterol | mmol/L | x | x |
| Triglycerides | mmol/L | x | x |
| HDL-C | mmol/L | x | x |
| LDL-C | mmol/L | x | x |

CAD: coronary artery disease. Includes history of myocardial infarction, percutaneous coronary intervention and coronary artery by-pass grafting. HDL-C: High density lipoprotein cholesterol; LDL-C: Low density lipoprotein cholesterol.

**Blood pressure.** Systolic and diastolic blood pressures (BP) are registered in mmHg at hospital admission and at both follow-up visits. Generally, at Swedish CR centres a blood pressure measurement is performed with the patient in supine position after five minutes of rest, either with an automated blood pressure monitor or sphygmomanometer and stethoscope.

**Blood samples.** Laboratory blood samples during hospitalization and both follow-up visits include fasting plasma glucose, total cholesterol, triglycerides and high-density lipoprotein cholesterol (HDL-C). Low-density lipoprotein cholesterol (LDL-C) is subsequently calculated using the Friedewald formula [17]. In the SWEDEHEART user manual it is recommended that lipids are measured according to local recommendations. Whether they are measured fasting or non-fasting is not registered in SWEDEHEART.

## Ethical considerations

At hospital admission, a nurse or physician verbally informs all patients eligible for inclusion in SWEDEHEART about the registry. All patients have the right to deny registration and the right upon request to be removed from the registry at any time. The Regional Ethical Review Board in Lund approved the current study (2014/6 and 2014/387).

## Statistical analysis

Baseline characteristics for attenders and non-attenders are presented as means ± standard deviations (SD) for normally distributed continuous variables, medians and interquartile ranges (IQR) for non-normally distributed continuous variables, and as percentages for categorical variables. Variable distribution was assessed by visual inspection of histograms and Q-Q plots and by calculating skewness and kurtosis. A z-value of between -1.69 and +1.69 was used to define normal distribution. To compare the demographics and risk factors of the groups at baseline, Student´s T-test, Mann-Whitney test (normally and non-normally distributed continuous variables) and Chi-square test (categorical variables) were used.

Linear and logistic multivariable regression analyses using a backward stepwise model were applied to study the associations between attending exCR and changes in the following risk

factor levels between baseline (measured during hospitalization) and the one-year follow-up visit: weight, BMI, systolic and diastolic BP, fasting plasma glucose, total cholesterol, triglycerides, HDL-C and LDL-C. Delta values between baseline and one-year follow-up rather than exact values at one-year were used as dependent variables to minimize the confounding effect of baseline differences between the groups. For smoking status, the variable from the one-year follow up was used, as smoking status is a dichotomous variable. As self-reported physical activity was not measured at baseline the actual one-year follow up variable was used (not delta). The following covariates were included in the models: from baseline data we included age, BMI (except for when BMI was the dependent variable), history of hypertension, previous CAD and heart failure, employment status, smoking (except for when smoking was the dependent variable); from one-year follow-up data we included use of statins, beta-blockers, ACE inhibitors and angiotensin II receptor blockers (ARB) and history of diabetes (self-reported or medication). All calculations were stratified by gender. Data was analysed by using the SPSS 22.0 statistical software package (IBM Corp. Armonk, NY). Significance level was set at $p < 0.05$ (2-tailed).

## Results

In total, 43% of the men and 44% of the women attended exCR (p = 0.100). Baseline characteristics are shown in Table 2. For both men and women, attenders were significantly younger (p<0.001 for both sexes). Further, there were differences in smoking status (never vs. previous or current smoker, p<0.001 for both sexes) and employment status (employed vs. retired or on sick leave, p<0.001 between groups for both sexes) between attenders and non-attenders and attenders had less comorbidities (diabetes (p<0.001 for both sexes), previous CAD/heart failure (p<0.001 for both sexes) and stroke (p = 0001 for men, <0.001 for women) and CAD medication (p<0.001 for all types and for both sexes–for details see Table 2) prior to the MI as compared to non-attenders. Attenders were on the other hand more often prescribed secondary preventive drugs at one-year follow-up compared to non-attenders (for details and p-values see Table 2). Dividing the cohort by gender and 5-year age bands, the highest attendance was observed among women 65 years and younger (approximately 50% in all groups), while the lowest attendance, 25%, was observed among the oldest women (≥76 years of age) (Fig 2).

Unadjusted differences (delta) in levels of risk factors between baseline and one-year follow-up as well as results of the multivariable regression analysis can be seen in Table 3. For both men and women, those who attended exCR were significantly more often reported having stopped smoking after the MI (current smokers at baseline only), reported more days per week of physical activity and a achieved a slightly larger reduction in triglycerides at one-year compared to non-attenders. Men attending exCR gained less weight while women attenders achieved a larger decrease in total cholesterol and LDL-C compared to non-attenders. There was no difference in systolic or diastolic BP, glucose and HDL-C between attenders and non-attenders, in men or in women.

## Discussion

In the present study, performed in the contemporary era when statins, ACE inhibitors and PCI have become a standard part of MI therapy, both men and women attending exCR achieved significantly larger improvements in cardiovascular risk factors between admission and one-year follow-up compared to non-attenders. For both sexes this included smokers more frequently reporting to have stopped smoking after the MI, attenders reporting a higher level of physical activity per week and achieving slightly larger reductions in triglycerides.

**Table 2. Baseline characteristics.**

| Attending exCR | Men (n = 14 312) | | | Women (n = 4 824) | | |
|---|---|---|---|---|---|---|
| | No | Yes | p-value | No | Yes | p-value |
| Number | 8161 | 6151 | | 2689 | 2135 | |
| Age (years), median (IQR) | 64 (12) | 64 (12) | <0.001[a] | 67 (11) | 64 (11) | <0.001[a] |
| Weight (kg), median (IQR) | 85 (17) | 85 (17) | 0.3[a] | 72 (19) | 72 (16) | 0.7[a] |
| BMI (kg/m$^2$), median (IQR) | 27.4 (5) | 27.2 (5) | 0.1[a] | 26.8 (7) | 26.6(6) | 0.2[a] |
| Employment status (%) | | | <0.001[c] | | | <0.001[c] |
| • Employed | 43.6 | 48.3 | | 25.6 | 39.9 | |
| • Retired | 51.3 | 47.1 | | 68.2 | 54.7 | |
| • Sick leave | 2.3 | 1.7 | | 4.0 | 3.2 | |
| • Unemployed, student, other | 2.8 | 2.9 | | 2.2 | 2.2 | |
| Smoking status (%) | | | <0.001[c] | | | <0.001[c] |
| • Never smoked | 31.3 | 36.6 | | 32.1 | 36.9 | |
| • Previous smoker > 1month | 38.4 | 40.5 | | 30.3 | 31.8 | |
| • Current smoker | 30.4 | 22.9 | | 37.6 | 31.3 | |
| Moist snuff (dipping tobacco) user (%) | 14.8 | 14.6 | 0.3[c] | 1.3 | 1.5 | 0.8[c] |
| Disease history (%) | | | | | | |
| • Diabetes | 19.5 | 14.9 | <0.001[c] | 22.7 | 16.3 | <0.001[c] |
| • Hypertension | 43.1 | 42.2 | 0.3[c] | 51.1 | 47.6 | 0.02[c] |
| • CAD/heart failure | 25.9 | 17.0 | <0.001[c] | 19.9 | 13.2 | <0.001[c] |
| • Stroke | 4.6 | 3.5 | 0.001[c] | 5.7 | 3.0 | <0.001[c] |
| Blood pressure (mmHg), median (IQR) | | | | | | |
| • Systolic | 150 (40) | 150 (39) | 1.0[a] | 150 (41) | 150 (40) | 0.6[a] |
| • Diastolic | 89 (21) | 89 (21) | 0.6[a] | 85 (20) | 85 (21) | 0.6[a] |
| Blood samples (mmol/L), median (IQR) | | | | | | |
| • Glucose, mean (SD) | 6.8 (3.0) | 6.8 (3.0) | 0.1[a] | 6.9 (3.0) | 6.7 (2.0) | 0.001[a] |
| • Total cholesterol, mean (SD) | 5.0 (1.7) | 5.1 (1.6) | 0.001[a] | 5.2 (1.8) | 5.4 (1.7) | <0.001[a] |
| • Triglycerides, median (q1, q3) | 1.4 (1.0) | 1.4 (1.0) | 0.049[a] | 1.40 (1.0) | 1.30 (1.0) | <0.001[a] |
| • HDL-C, mean (SD) | 1.1 (0.4) | 1.1 (0.4) | 0.003[a] | 1.3 (0.5) | 1.3 (0.5) | <0.001[a] |
| • LDL-C, mean (SD) | 3.1 (1.5) | 3.2 (1.5) | <0.001[a] | 3.1 (1.6) | 3.3 (1.6) | <0.001[a] |
| Prior medication (%) | | | | | | |
| • ACE or ARB | 34.0 | 31.1 | <0.001[c] | 35.7 | 30.6 | <0.001[c] |
| • ASA | 30.3 | 23.1 | <0.001[c] | 29.4 | 22.4 | <0.001[c] |
| • Beta-blockers | 30.4 | 24.8 | <0.001[c] | 33.7 | 28.3 | <0.001[c] |
| • Statins | 30.1 | 26.1 | <0.001[c] | 29.2 | 23.1 | <0.001[c] |
| Infarction type (%) | | | <0.001[c] | | | <0.001[c] |
| • STEMI | 37.1 | 42.0 | | 32.0 | 36.8 | |
| • NSTEMI | 62.9 | 58.0 | | 68.0 | 63.2 | |
| Medication at one-year post MI (%) | | | | | | |
| • ACE or ARB | 80.8 | 83.1 | <0.001[c] | 76.3 | 78.7 | 0.04[c] |
| • ASA | 91.5 | 93.0 | 0.004[c] | 88.5 | 91.3 | 0.002[c] |
| • Beta-blockers | 85.0 | 87.2 | <0.001[c] | 84.2 | 86.0 | 0.03[c] |
| • Statins | 91.8 | 94.4 | <0.001[c] | 86.1 | 89.7 | <0.001[c] |

IQR: interquartile range; SD: standard deviation; BMI: body mass index; CAD: Coronary Artery Disease [myocardial infarction, percutaneous coronary intervention, coronary artery bypass surgery]; HDL-C: High density lipoprotein cholesterol; LDL-C: Low density lipoprotein cholesterol; ACE: Angiotensin converting enzyme; ARB: Angiotensin receptor blocker; ASA: acetyl salicylic acid; STEMI: ST elevation myocardial infarction; NSTEMI: Non-ST elevation myocardial infarction.

[a] Mann-Whitney U test

[b] Student´s t-test

[c] Chi-2 test.

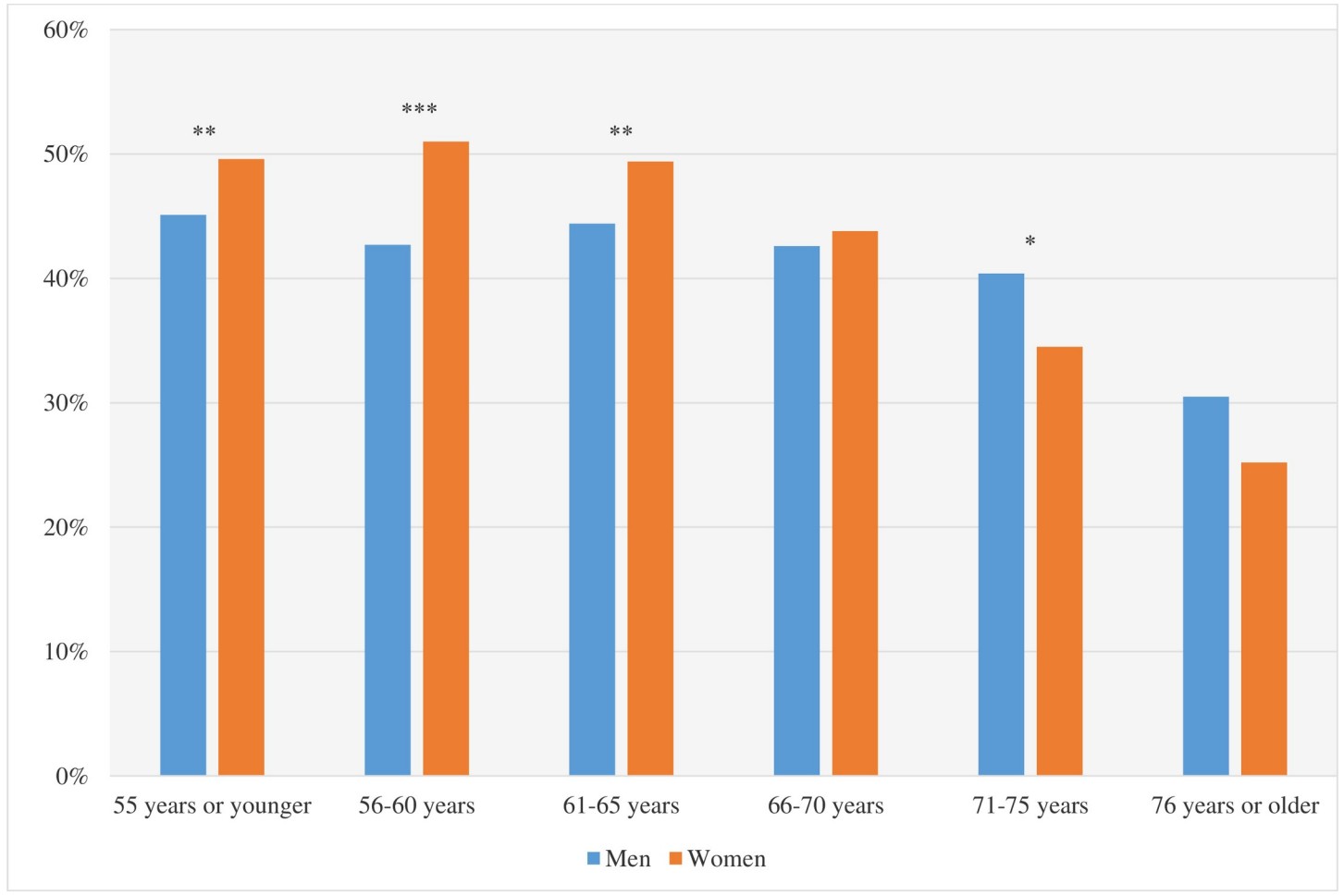

**Fig 2. Proportion of patients attending exCR stratified by age groups.** The figure shows the proportion of patients attending exCR during the first-year post MI, stratified by gender and age groups. The lowest attendance was observed among the oldest women (≥76 years). *p<0.05; **p<0.01; ***p<0.001.

Among men we also observed less weight gain whereas among women we observed larger improvements in total cholesterol and LDL-C in attenders compared to non-attenders.

Attending exCR has been shown to positively affect cardiovascular risk factors in several patient groups at high cardiovascular risk, demonstrating significant improvements in weight reduction, BP and lipid and glycaemic control [18–22]. Being surrogate endpoints, however, one can speculate to the relevance of changes in risk factors on long-term outcomes. The clinical value of changes in physical activity levels has been indicated in several studies [23–25]. In the study by Ekblom et al, which was based on SWEDEHEART data using the same self-report instrument as in our study, a 14% reduction in mortality risk was observed for each additional day per week patients reported being physically active [23]. The difference between attenders and non-attenders in our study was 0.5 days for men and 0.8 days for women, which would amount to 7% and 12% mortality risk reduction, respectively. As patients´ previous physical activity levels were unknown, we cannot claim that participation in exCR was the only factor explaining the difference in physical activity levels at follow-up. In contrast to the well-known effects of aerobic exercise capacity as a strong predictor of mortality in patients with CAD [26–28], the impact of increased physical activity levels per se has not yet been clarified and needs to be further studied. Also, the previously cited studies [23–25] as well as the present study

**Table 3. Differences in cardiovascular risk factor changes between exCR attenders and non-attenders at one-year post MI.**

| Attending exCR | Men | | | Women | | |
|---|---|---|---|---|---|---|
| | No | Yes | p-value | No | Yes | p-value |
| N | 8161 | 6151 | | 2689 | 2135 | |
| Weight (kg), mean (SD) | +0.3 (5.7) | +0.0 (5.7) | 0.01[a] | +0.7 (5.8) | +0.4 (5.9) | 0.2[a] |
| BMI (kg/m$^2$), mean (SD) | +0.1 (1.8) | +0.0 (1.8) | 0.02[a] | +0.3 (2.2) | +0.2 (2.2) | 0.2[a] |
| Blood pressure (mmHg), mean (SD) | | | | | | |
| • Systolic | -18.1(28.9) | -20.1 (28.6) | 0.05[a] | -18.0 (31.7) | -19.5 (30.3) | 0.8[a] |
| • Diastolic | -10.4 (17.5) | -11.0 (17.1) | 0.9[a] | -9.2 (18.4) | -9.3 (17.4) | 0.4[a] |
| Proportion of current smokers at baseline reporting at one-year to have stopped smoking after the MI[c] | 50% | 64% | <0.001[b] | 53% | 64% | <0.001[b] |
| Glucose (mmol/L), mean (SD) | -1.3 (3.1) | -1.5 (2.9) | 0.7[a] | -1.6 (3.6) | -1.6 (3.3) | 1.0[a] |
| Total cholesterol (mmol/L), mean (SD) | -1.0 (1.3) | -1.2 (1.3) | 0.5[a] | -0.9 (1.4) | -1.2 (1.4) | <0.001[a] |
| Triglycerides (mmol/L), mean (SD) | -0.1 (0.9) | -0.2 (0.8) | 0.001[a] | -0.0 (0.8) | -0.1 (0.6) | 0.01[a] |
| HDL-C (mmol/L), mean (SD) | +0.1 (0.3) | +0.1 (0.3) | 0.3[a] | +0.1 (0.3) | +0.1 (0.3) | 0.5[a] |
| LDL-C (mmol/L), mean (SD) | -1.0 (1.1) | -1.1 (1.1) | 0.7[a] | -0.9 (1.2) | -1.2 (1.2) | <0.001[a] |
| Days of self-reported PA/week, mean (SD)[c] | 3.4 (2.7) | 3.9 (2.5) | <0.001[a] | 3.0 (2.8) | 3.8 (2.6) | <0.001[a] |

Adjusted for: age, BMI, employment status, smoking, hypertension and CAD/heart failure at baseline and use of statins, ACE/ARB and history of diabetes at one-year follow-up. SD: standard deviation; BMI: Body mass index; CAD; coronary artery disease; HDL: High density lipoprotein; LDL: Low density lipoprotein; PA: physical activity.

[a]Multiple linear regression

[b]Multiple logistic regression

[c]Actual status at one year follow-up.

were limited to self-reported physical activity, entailing a risk of over- or under-estimation, as well as issues of recall and response bias [29]. Therefore, in future studies using objective measured physical activity in terms of accelerometry is suggested [30].

For both men and women attenders smokers more frequently reporting to have stopped smoking after the MI compared to non-attenders. In a meta-analysis including 20 studies on smoking cessation in post MI patients smoking cessation resulted in a 36% relative risk reduction for total mortality [31]. With the differences in smoking cessation rates between attenders and non-attenders in our study being quite large (64% vs 50% and 53% for men and women, respectively), the clinical relevance of the results can be assumed to be substantial.

We observed a 0.3 mmol/l difference in LDL reduction between female attenders and non-attenders in our study. In the most recent meta-analysis from the Cholesterol Treatment Trialists´ Collaboration a 1 mmol/L reduction in LDL was projected to result in a 21% reduction in mortality risk [32]. These estimates would translate to approximately 7% future mortality risk reduction for women in our study. The small difference observed in triglyceride levels is on the other hand probably of limited clinical significance.

Taylor et al [33] used the IMPACT model to investigate the extent to which the reduction in cardiovascular disease mortality seen among CAD patients attending exCR was attributable to changes in risk factors. The study showed that approximately half of the 28% reduction in cardiovascular disease mortality achieved with attending an exCR programme was attributed to improvements in risk factors measured in the study. Our results from an unselected real-life MI patient population, with high nationwide coverage, contribute to this pool of evidence.

There are several other reasons why patients who attend exCR might improve their prognosis. In a study by Bäck et al [34], a larger improvement in maximal aerobic capacity was

observed in patients after elective PCI randomized to high frequency exercise intervention as an addition to traditional CR. Aerobic exercise capacity is a strong predictor of mortality in patients with CAD, and therefore a small gain in maximal aerobic capacity may improve not only functional capacity but also survival prospects [26–28]. Also, exercise training releases endorphins, which increase the feeling of well-being and might encourage the patient to continue exercising and keep away from unhealthy habits [35]. Attending group-based exCR, meeting peers with the same diagnosis, under the supervision of trained physiotherapists, might make patients feel more secure and motivated to exercise at a higher level [36]. Also, patients might inspire each other to a healthier lifestyle such as being smoke-free and eating healthier foods, subjects frequently discussed in the exCR groups. Finally, peer support has been shown to positively affect cardiovascular risk factors [37]. Indeed, this might explain the difference between attenders and non-attenders reporting to have stopped smoking after the MI at the one-year follow-up visit in our study. Giallauria et al [38] showed in their study that extended attendance in a CR programme resulted in a higher likelihood of long-term smoking abstinence in patients after MI. As exCR constitutes a substantial part of a comprehensive CR programme this also might partly explain our findings.

Some differences in cardiovascular risk factor improvement between men and women were observed in our study. Improvement in weight was only observed in men while improvements in total cholesterol and LDL-C were on the other hand only observed in women. Lower referral rates for women compared to men is a well described problem within CR, contributing to worse goal attainment for lifestyle and risk factor targets for women compared to men [39–42]. Also, previous studies have shown that women more frequently drop out of CR programmes [10]. While the overall attendance in exCR in our study was the same for men and women (43% vs 44%), the number of completed exCR programme sessions and adherence to the prescribed exercise regime was not documented in SWEDEHEART at the time of the study. Potential gender differences in these parameters could as such partly explain the differences in outcomes between men and women in our study. Also, the physiological response to exercise training has been shown to be different for men and women [43], which might explain some of the observed gender differences in risk factor improvement in our study.

Additionally, we found that men and women who attended exCR were younger, had less comorbidities and cardiovascular risk factors than non-attenders. This is in line with previous studies on CR attendance [9, 41, 42]. Reasons for patients not attending CR in general, and exCR in particular, are discussed in several studies. In a recently published study, Borg et al [44] found that distance to the CR-centre was the strongest predictor for non-attendance in exCR. Additional significant factors included smoking, a higher burden of comorbidities, and male sex. Cooper et al [45] assessed patients' beliefs about CR to better predict who were likely to attend. They found that patients who attended believed that CR was necessary and understood its role as a part of their treatment while non-attenders were more likely to believe that CR was more suitable for younger patients, expressed concerns about exercise training or reported practical barriers to attendance. These findings also correspond to the results found by McKee G et al [46] and by Mikkelsen T et al [47] where lack of interest, that the programme would not be beneficial or lack of time due to work were common reasons why patients were not attending exCR. As observed in previous studies we found the lowest attendance among women in the highest age-groups (71–75 and ≥76 years) [9, 10, 42]. At the same time, attendance in the whole cohort was the same for men and women. A possible explanation is the age limit of <75 years for obligatory registration of CR variables in SWEDEHEART, leading to under-registration of those ≥75 years and thereby fewer elderly women in our cohort, who generally have the lowest attendance [10, 42, 48]. Also, a higher proportion of the women (both attenders and non-attenders) were retired (p<0.001), and fewer had previous CAD or

heart failure (p<0.001), compared to men. Being retired and without significant comorbidities might explain better attendance rates for women in our cohort compared to that reported in previous studies.

## Strengths and weaknesses

In this study, we analysed a large data set from the SWEDEHEART registry–an internationally unique database on patients post MI from a real-life setting with high national coverage [13, 14]. Data on attending exCR in real-life as opposed to that from controlled trials is of great value, as internal validity can be affected when studying effects of exercise training using a controlled study design, creating the so-called testing phenomenon. This happens when participants in a "testing" environment perform due to expectations as opposed to how they perform in every-day life. Other strengths are the population being more representative of usual patients with MI in terms of age, gender, risk factors and comorbidities than patients in randomized controlled trials [3].

There are limitations to using registry-based data. Being an observational study, no conclusions can be drawn about cause-effect relationships. Several variables were self-reported, which might have affected data validity. Also, the selection process was non-randomized. To overcome the difference between the groups at baseline we chose to analyse changes in risk factor levels between admission or two-month follow-up and one-year follow-up data instead of actual values at the one-year follow-up.

When the study was conducted there was not yet a clear definition in the register on how to define a successfully completed exCR programme. Even though the total duration of the programme was recommended to be at least three months, patients could still answer yes to this variable if they have not attended a full three-month programme. Therefore, we cannot report on number of centre-based or home-based exercise sessions, and as such this can vary considerably between patients. The SWEDEHEART register has recently (2016) added new registry outcomes of aerobic capacity and muscle strength, making it possible to measure changes in physical fitness in future studies.

Finally, those attending exCR might also have had better attendance in other CR programme elements that have been shown to have beneficial effects on risk factor management and long-term outcomes after CR, such as patient education [49], causing a potential bias in our findings. Residual confounding by degree of motivation for behavioural change, socioeconomic status, personality and social interaction is also likely.

## Conclusions

In conclusion, in a large cohort of patients with MI we found that patients attending exCR achieved larger improvements in cardiovascular risk factors, including higher rates of smoking cessation, increased self-reported physical activity, and a larger reduction in triglyceride levels (both sexes), weight control (men) and cholesterol levels (women). Our observations align with previous findings from controlled trials on the benefits of exCR on future cardiovascular disease risk, for both men and women, reinforcing the previously identified importance of exCR as a part of comprehensive CR [6, 50]. Also, our results support that exercise training maintains an important place in secondary prevention post MI even in the era of modern therapy. Additionally, we observed that those attending exCR were younger, had less comorbidities, and less cardiovascular risk factors at baseline compared to non-attenders. How exercise training differently affects men and women and how attendance of older and frailer patients can be improved needs to be addressed.

## Acknowledgments

The authors are especially thankful to all the patients who have contributed their data to the SWEDEHEART registry and the hospital and CR personnel who enter data to the registry every day.

## Author Contributions

**Conceptualization:** Ingela Sjölin, Maria Bäck, Lennart Nilsson, Margret Leosdottir.

**Data curation:** Ingela Sjölin, Margret Leosdottir.

**Formal analysis:** Ingela Sjölin, Margret Leosdottir.

**Methodology:** Ingela Sjölin, Maria Bäck, Lennart Nilsson, Margret Leosdottir.

**Project administration:** Alexandru Schiopu, Margret Leosdottir.

**Resources:** Margret Leosdottir.

**Software:** Margret Leosdottir.

**Supervision:** Maria Bäck, Lennart Nilsson, Alexandru Schiopu, Margret Leosdottir.

**Validation:** Margret Leosdottir.

**Visualization:** Ingela Sjölin, Maria Bäck, Alexandru Schiopu.

**Writing – original draft:** Ingela Sjölin.

**Writing – review & editing:** Maria Bäck, Lennart Nilsson, Alexandru Schiopu, Margret Leosdottir.

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
