## [Decision Letter · Decision Letter 0]

17 Dec 2019

PONE-D-19-32783

Association between attending exercise-based cardiac rehabilitation and cardiovascular risk factors at one-year post myocardial infarction

PLOS ONE

Dear Dr leosdottir,

Thank you for submitting your manuscript to PLOS ONE. After careful consideration, we feel that it has merit but does not fully meet PLOS ONE’s publication criteria as it currently stands. Therefore, we invite you to submit a revised version of the manuscript that addresses the points raised during the review process.

Please address comments from all reviewers

We would appreciate receiving your revised manuscript by Jan 31 2020 11:59PM. To enhance the reproducibility of your results, we recommend that if applicable you deposit your laboratory protocols in protocols.io, where a protocol can be assigned its own identifier (DOI) such that it can be cited independently in the future. For instructions see: http://journals.plos.org/plosone/s/submission-guidelines#loc-laboratory-protocols

We look forward to receiving your revised manuscript.

Kind regards,

Gordon McGregor

Academic Editor

PLOS ONE

Journal Requirements:

1. We note that you have indicated that data from this study are available upon request. PLOS only allows data to be available upon request if there are legal or ethical restrictions on sharing data publicly. For information on unacceptable data access restrictions, please see http://journals.plos.org/plosone/s/data-availability#loc-unacceptable-data-access-restrictions.

Reviewers' comments:

Reviewer's Responses to Questions

**Comments to the Author**

1. Is the manuscript technically sound, and do the data support the conclusions?

Reviewer #1: Yes

Reviewer #2: Yes

Reviewer #3: Yes

Reviewer #4: Yes

2. Has the statistical analysis been performed appropriately and rigorously? 

Reviewer #1: Yes

Reviewer #2: Yes

Reviewer #3: I Don't Know

Reviewer #4: Yes

3. Have the authors made all data underlying the findings in their manuscript fully available?

Reviewer #1: Yes

Reviewer #2: Yes

Reviewer #3: Yes

Reviewer #4: Yes

4. Is the manuscript presented in an intelligible fashion and written in standard English?

Reviewer #1: Yes

Reviewer #2: Yes

Reviewer #3: Yes

Reviewer #4: Yes

5. Review Comments to the Author

Reviewer #1: It is always good to see actual audit type data against which to compare data collected from studies. The paper is presented in a logical easy to read and clear format, though a number of considerations are required to get it to publishable standard. Generally the Discussion was descriptive and needs more intra and inter-study critical analyses and comparisons.

How do the data compare with the UK's NACR and the US AACVPR data sets/registry's and other registry's from Europe?

As this study has a strong focus on physical activity, in making comparisons with the evidence, a key element which makes the findings weak is that the outcomes have only been linked with frequency of physical activity. Do the SWEDEHeart programmes not measure fitness and exercise capacity? Aerobic fitness is a very strong risk factor; far stronger than physical activity. The discussion alludes to changes in aerobic fitness, so if you do not have these measures it is important to highlight this weakness and make recommendations for future Registry Outcomes. Is there in any information in the Data sets to whether or not patients actually performed the 20+ minutes at an RPE >13?

Another key weakness which needs to be noted is the single use of Self-Reported Physical activity. Remarks in the Discussion should be made on the validity and reliability of this method. The final point on physical activity, is that although the differences between attenders and non-attenders participation levels were statistically significant (presumably due to the large sample size), is a 0.5 day difference actually clinically meaningful? This small real difference, though statistically significant, could likely be a function the limitations of Self Report, which are known to greatly over-report in Physical Activity (see reports from S. Prince from Statistics Canada) Again, the benefit of this difference would have been illuminated if exercise capacity were measured as a risk factor outcome.

In terms of manuscript presentation clarity and noting the main results are summarised in Tables 4 and 5., the titles of these tables seem a little misleading and also need more precision. The term "association" would lead one to believe a correlation statistic was being used and not a test of "difference" being assessed. I would recommend that these tables are renamed with such recommended words as; Differences in CVD risk factors 12-months after treatment of a CHD event, in those who attended versus those who declined cardiac rehabilitation.

Reviewer #2: This study appears to be methodologically sound, and generates interesting results, particularly around the gender differences. The only critique that I have is around the fact that there is no data around adherence to / % completion of the programme - I do think that would be interesting to know, especially given that women have a higher attendance than men (which is in contrast to what's found typically in other countries).

Reviewer #3: The authors present an interesting report on effects of exercise-based CR on cardiovascular risk factors. The retrospectively analysed dataset is large and includes a non-randomised control group. The data is timely and provides valuable data that reports on the effects of cardiac rehabilitation in a ‘real world’ setting. The manuscript would benefit from some modifications and/or clarifications.

Abstract

Well written – The authors may wish to revise some of the conclusions based on my comments, although I will leave this for them to decide.

Background

The background and rationale is well articulated.

Line 62 – I agree that the effect of exercise-based CR in the modern era of MI treatment is less well document. However, the systematic review and meta-analysis by Powell et al. (2018) investigated this topic in relation to mortality/morbidity and found no effect. Please could you briefly acknowledge the findings of ‘Is exercise-based cardiac rehabilitation effective? A systematic review and meta-analysis to re-examine the evidence’.

Methods

Line 101 – Could patients who don’t complete a full cardiac rehabilitation programme be included in this analysis? For example, could someone who undertakes 2 months of rehabilitation be recorded as someone who attended cardiac rehabilitation? (yes/no). Equally, is it likely that many patients had more than 3 months of cardiac rehabilitation, and therefore, had a greater treatment ‘dose’.

Table 1 – Please clarify if lipids were measured from fasting/non-fasting blood samples.

Line 121 – The use of CHD and CAD are both used in the manuscript. Please amend use one throughout the manuscript for consistency.

Line 139 – pleased change to ‘reported)’. Also, please amend to ‘otherwise it is self-reported).

143 – Are you able to say whether BP measurements are generally obtained from automated or sphygmomanometer and stethoscope?

Line 159 – This sentence is a little confusing. Please can you clarify that means and SD were used to report normally distributed data, and medians with IQR were used to report non-normally distributed data. Please can you also clarify how data normality were assessed. Please change ‘skewed ’to read ‘non-normally distributed’ or ‘data that was not normally distributed because assessing kurtosis is also part of determining normality.

Line 175 – Please can you justify why beta-blockers weren’t included as a covariate? They commonly have an effect on blood pressure as well as resting heart rate.

General comment – It isn’t clear from the methods, why women and men were analysed separately. It would be useful to report differences between all four conditions; male attenders, male non-attenders, female attenders, and female non-attenders. The comments below relate to this point.

Results

Table 2 – It would be helpful if the descriptive statistics for men and women were presented in one table.

Table 2 – Please explain what wet snuff (%) is. Many readers may not be familiar with this.

Line 212 – See comments above. Would it be more informative to compare males and females if these data were analysed as four different groups, and reported in one table?

215 – Please explain what is meant by being ‘abstinent’ from smoking. This isn’t defined in the methods. Smoking status is reported differently in Tables 4 and 5, compared with Tables 2 an 3. Until these variables are consistently reported, it makes the conclusions around the smoking status difficult to justify as the data could be misleading.

Discussion

The claims surrounding the statistically significant reduction in risk factors is fine. However, some of the significant reductions are small e.g. 0.1 mmol/ greater reduction in triglycerides among women taking part in CR. What is the clinical value of this, and how does this compare to data from clinical trials? This would be valuable information to reader because the aim of the study is to examine whether exercise-based CR results in beneficial changes in cardiovascular risk factors in a ‘real life’ setting.

Without knowing patient physical activity levels at baseline, it is difficult to claim the exercise-based CR increased PA levels.

Line 261 to 272– This section is quite off topic because the discussion surrounding mortality benefit isn’t focused on the observed significant improvements in cardiovascular risk factors. It would be more valuable to discuss the improvements observed in this trial and how this might improve survival/morbidity. This would be particularly interesting if the changes in cardiovascular risk factors are comparable to those reported in other studies.

Line 289 – This line relates to the point I highlighted earlier regarding data on how long patients took part in CR for. Please can you discuss this within the methods or results as appropriate.

Line 327 – This shouldn’t be a new paragraph if starting with ‘on the other hand’.

Reviewer #4: This is a really important paper and I have only very minor comments regarding the clarity of difference between exCR and CR in the Patient population settings. Can you please explain how it was possible to say the changes were attributable to exercise and not to the other elements of the comprehensive CR progamme?

6. PLOS authors have the option to publish the peer review history of their article (what does this mean?). If published, this will include your full peer review and any attached files.

Reviewer #1: No

Reviewer #2: No

Reviewer #3: No

Reviewer #4: No

---

## [Author Response · Author response to Decision Letter 0]

6 Feb 2020

We thank the Reviewers for their valuable and constructive comments. A Point-to-Point list of answers can be found in the uploaded file Response to Reviewers.

---

## [Decision Letter · Decision Letter 1]

11 Mar 2020

PONE-D-19-32783R1

Association between attending exercise-based cardiac rehabilitation and cardiovascular risk factors at one-year post myocardial infarction

PLOS ONE

Dear Dr leosdottir,

Thank you for submitting your manuscript to PLOS ONE. After careful consideration, we feel that it has merit but does not fully meet PLOS ONE’s publication criteria as it currently stands. Therefore, we invite you to submit a revised version of the manuscript that addresses the points raised during the review process.

Please respond to the minor points raised by reviewer 2

We would appreciate receiving your revised manuscript by Apr 25 2020 11:59PM. To enhance the reproducibility of your results, we recommend that if applicable you deposit your laboratory protocols in protocols.io, where a protocol can be assigned its own identifier (DOI) such that it can be cited independently in the future. For instructions see: http://journals.plos.org/plosone/s/submission-guidelines#loc-laboratory-protocols

We look forward to receiving your revised manuscript.

Kind regards,

Gordon McGregor

Academic Editor

PLOS ONE

Reviewers' comments:

Reviewer's Responses to Questions

**Comments to the Author**

1. If the authors have adequately addressed your comments raised in a previous round of review and you feel that this manuscript is now acceptable for publication, you may indicate that here to bypass the “Comments to the Author” section, enter your conflict of interest statement in the “Confidential to Editor” section, and submit your "Accept" recommendation.

Reviewer #2: (No Response)

Reviewer #3: All comments have been addressed

Reviewer #4: All comments have been addressed

2. Is the manuscript technically sound, and do the data support the conclusions?

Reviewer #2: Yes

Reviewer #3: Yes

Reviewer #4: Yes

3. Has the statistical analysis been performed appropriately and rigorously? 

Reviewer #2: Yes

Reviewer #3: Yes

Reviewer #4: Yes

4. Have the authors made all data underlying the findings in their manuscript fully available?

Reviewer #2: Yes

Reviewer #3: No

Reviewer #4: Yes

5. Is the manuscript presented in an intelligible fashion and written in standard English?

Reviewer #2: Yes

Reviewer #3: Yes

Reviewer #4: (No Response)

6. Review Comments to the Author

Reviewer #2: This is a well written article which helps argue the positive impact of CR upon CVD risk factors (particularly useful given that the effectiveness of CR has been questioned within the literature recently).

There are only minor revisions required:

line 43 - significantly larger than? ('non-attenders' needs added here)

lines 194-204 - it would be helpful to have the p values after each finding here

line/table 206 - personally I find the placement of the p values confusing, e.g. why is the p value for employment status only aligned with employment status when the text states that that the difference lies between those who are employed and those who aren't? I think the table could be clearer

lines 285-300 - here it would be good to refer back to reference [8] (as a quite prominent article presenting negative results in relation to CR) and discuss your findings in relation to this

This is another good reference / notable paper which would be good to discuss in relation to your results:

Van Halewijn G, Deckers J, Tay HY, van Domburg R, Kotseva K & Wood D. Lessons from contemporary trials of cardiovascular prevention and rehabilitation. International Journal of Cardiology 2017; 232:294-303

Reviewer #3: Thank your for taking the time to accommodate the recommendations. I am happy with the work that you have done and look forward to seeing this manuscript published.

Reviewer #4: (No Response)

7. PLOS authors have the option to publish the peer review history of their article (what does this mean?). If published, this will include your full peer review and any attached files.

Reviewer #2: No

Reviewer #3: No

Reviewer #4: No

---

## [Author Response · Author response to Decision Letter 1]

20 Apr 2020

Please see attached Word document titled Response to reviewers.

---

## [Editor Report · Decision Letter 2]

22 Apr 2020

Association between attending exercise-based cardiac rehabilitation and cardiovascular risk factors at one-year post myocardial infarction

PONE-D-19-32783R2

Dear Dr. leosdottir,

We are pleased to inform you that your manuscript has been judged scientifically suitable for publication and will be formally accepted for publication once it complies with all outstanding technical requirements.

With kind regards,

Gordon McGregor

Academic Editor

PLOS ONE
---

## [Editor Report · Acceptance letter]

28 Apr 2020

PONE-D-19-32783R2 

Association between attending exercise-based cardiac rehabilitation and cardiovascular risk factors at one-year post myocardial infarction 

Dear Dr. leosdottir:

I am pleased to inform you that your manuscript has been deemed suitable for publication in PLOS ONE. Congratulations! Your manuscript is now with our production department. 

With kind regards,

on behalf of

Dr. Gordon McGregor 

Academic Editor

PLOS ONE